

# Evaluation of phytochemical profile, and antioxidant, antidiabetic activities of indigenous Thai fruits

Jirayupan Prakulanon[1], Sutsawat Duangsrisai[1], Srunya Vajrodaya[1] and Thanawat Thongchin[2]

[1] Department of Botany, Kasetsart University, Bangkok, Thailand
[2] Department of Medical Science, Ministry of Public Health, Medicinal Plant Research Institute, Nonthaburi, Thailand

Corresponding author
Sutsawat Duangsrisai,
fscissw@ku.ac.th

## ABSTRACT

**Background**. This research aims to explore the phenolics identification, phenolics quantification, antioxidant and potential biofunctional properties of lesser-known Thai fruits and their potency to treat type 2 diabetes mellitus (T2DM). Including, *Antidesma puncticulatum, Dillenia indica, Diospyros decandra, Elaeagnus latifolia, Flacourtia indica, Garcinia dulcis, Lepisanthes fruticose, Mimusops elengi, Muntingia calabura, Phyllanthus reticulatus, Streblus asper, Syzygium cumini, Syzygium malaccense, Willughbeia edulis* and *Schleichera oleosa* were analyzed by their phenolic and flavonoid content. These fruits have received limited scientific attention, prompting an investigation into their health benefits, particularly their relevance to diabetes management.

**Methods**. The study utilized methanolic crude extracts to measure phenolic and flavonoid levels. Additionally, UHPLC-DAD was utilized to identify and quantify phenolics. The methanolic extracts were assessed for antioxidant and antidiabetic abilities, including $\alpha$-glucosidase and $\alpha$-amylase inhibition.

**Results and Conclusion**. The study highlighted *S. cumini* as a rich source of phenolic (980.42 $\pm$ 0.89 mg GAE/g and flavonoid (3.55 $\pm$ 0.02 mg QE/g) compounds with strong antioxidant activity (IC$_{50}$ by DPPH; 3.00 $\pm$ 0.01 $\mu$g/ml, IC$_{50}$ by ABTS; 40 $\pm$ 0.01 $\mu$g/ml, FRAP; 898.63 $\pm$ 0.02 mg TE/ml). Additionally, *S. cumini* exhibited promising antidiabetic effects (*S. cumini* IC$_{50}$; 0.13 $\pm$ 0.01 mg/ml for $\alpha$-glucosidase inhibition, 3.91 $\pm$ 0.05 mg/ml for $\alpha$-amylase inhibition), compared to Acarbose (IC$_{50}$; 0.86 $\pm$ 0.01 mg/ml for $\alpha$-glucosidase inhibition, 0.39 $\pm$ 0.05 mg/ml for $\alpha$-amylase inhibition). Remarkably, compounds like catechins, gallic acid, kaempferol, and ellagic acid were identified in various quantities. This study suggests that these fruits, packed with phenolics, hold the potential to be included in an anti-diabetic diet and even pharmaceutical applications due to their health-promoting properties.

## INTRODUCTION

Diabetes mellitus (DM) is a serious chronic non-communicable disease that has seen a dramatic increase in prevalence in the past three decades. According to the World Health Organization (WHO), around 422 million people worldwide have diabetes, with the

majority living in low-middle-income countries. Diabetes is characterized by high blood glucose levels, which can damage the heart, blood vessels, eyes, kidneys, and nerves (*WHO, 2016*). There are two types of diabetes. Type 1 is caused by β-cell destruction and absolute insulin deficiency since birth, and type 2 is the most common form and is associated with overweight and obesity, characterized by various degrees of β-cell dysfunction and insulin resistance. Type 2 diabetes can be prevented through healthy lifestyle choices such as regular exercise, avoiding smoking, and eating a healthy diet (*Roglic, 2016*). Currently, there are various pharmacological approaches to prevent and treat DM. Antioxidant agents and lifestyle changes to adjust to a healthy diet are most common. Whereas taking oral hypoglycemic drugs, which inhibit carbohydrate digestion enzymes such as α-glucosidase and α-amylase, are proven effective in preventing DM but often have accompanying side effects (*Proença et al., 2021*).

Efforts to manage and prevent diabetes have led to a multifaceted approach, encompassing pharmacological interventions and lifestyle modifications. Among these strategies, the role of antioxidants, particularly those found in natural sources like fruits, has garnered considerable attention. Antioxidants, particularly phenolics, are crucial in safeguarding and sustaining the body against diabetes. They work by preventing radical-induced damage to β-cells, which, if unchecked, can lead to β-cell failure and subsequently result in diabetes. Moreover, these agents contribute to maintaining optimal oxidant levels within β-cells, thereby reducing oxidative stress (*Kaneto et al., 1999*). Phenolic compounds have been reported to inhibit radicals through mechanisms such as hydrogen atom transfer, transfer of a single electron, sequential proton loss electron transfer, and chelation of transition metals. The hydroxyl group and benzene ring in their structure play crucial roles. The hydroxyl group functions in antioxidation by donating electrons to radicals, while the benzene ring stabilizes antioxidant molecules through reactions with free radicals (*Zeb, 2020*). Simultaneously, diabetes management can involve inhibiting enzymes such as α-amylase, which breaks down complex carbohydrates into smaller polysaccharides, and α-glucosidase, which breaks down disaccharides and oligosaccharides into glucose. Glucose can ultimately be absorbed by the body. Furthermore, by inhibiting enzymes, glucose absorption can be slowed down, potentially aiding in controlling blood sugar levels (*Gong et al., 2020*). For example, kaempferol inhibits diabetes by boosting glucose metabolism in skeletal muscle and inhibiting gluconeogenesis in the liver (*Alkhalidy et al., 2018*), catechin alleviates hyperglycemia by enhancing insulin sensitivity, reducing oxidative stress and modulating mitochondrial function (*Wen et al., 2022*), ellagic acid lowers glucose and lipid levels through the inhibition of β-cell apoptosis and the stimulation of insulin production (*Harakeh et al., 2020*), and gallic acid has been reported to be found in high content in Indian gooseberry (*Phyllanthus emblica*) and has antioxidant and antidiabetic activities by reducing blood glucose levels (*Elobeid & Ahmed, 2015*; *Sawant, Pandita & Prabhakar, 2010*). Additionally, a previous study investigated Thai fruits' antioxidant and antidiabetic activities. Mangosteen (*Garcinia mangostana*) fruit peel and Indian gooseberry (*Phyllanthus emblica*) had high phenolic contents and antioxidant activities, whereas mulberry (*Morus alba*) had the strongest α-glucosidase inhibitory activity (*Nanasombat, Yansodthee & Jongjaited, 2018*). Brazilian peppertree (*Schinus terebinthifolius*) had a high

phenolic content, antioxidant activity and α-glycosidase inhibitory activity (*Dedvisitsakul & Watla-Iad, 2022*).

Thailand has a diverse range of fruits every season, yet many of them remain underexplored their phytochemical and biological properties. Therefore, this study aims to evaluate the phytochemical profiles focusing on phenolics, antioxidant potential, and antidiabetic potentials, particularly the inhibition of carbohydrate digestive enzymes, which can help manage blood glucose levels, are relevant in diabetes. Hence, the study aimed to investigate the *in-vitro* α-amylase and α-glucosidase inhibitory activity of 15 less-researched, selected local fruits in Thailand, focusing on their commonality, affordability, and accessibility. Including, *Antidesma puncticulatum*, *Dillenia indica*, *Diospyros decandra*, *Elaeagnus latifolia*, *Flacourtia indica*, *Garcinia dulcis*, *Lepisanthes fruticosa*, *Mimusops elengi*, *Muntingia calabura*, *Phyllanthus reticulatus Streblus asper*, *Syzygium cumini*, *Syzygium malaccense*, *Schleichera oleosa* and *Willughbeia edulis*. We anticipate these findings will provide valuable groundwork for future research on these indigenous Thai fruits' antioxidant and antidiabetic properties.

## MATERIALS & METHODS

### Chemicals and reagents

The following chemicals were used in the experiments: HPLC grade water containing 0.1% $H_2PO_4$ and methanol containing 0.1% $H_2PO_4$ (Phosphoric acid) were used for the HPLC analysis and purchased from Merck (Darmstadt, Germany). Standard HPLC grade, including catechin, ellagic acid, epicatechin, epicatechin gallate, gallic acid, and kaempferol were purchased from Sigma-Aldrich (St. Louis, MO, USA). The Folin-Ciocalteu phenol reagent was purchased from Merck (Darmstadt, Germany). Methanol A.R. was purchased from RCL Labscan (Ireland). 2,2-diphenyl-1-picrylhydrazyl (DPPH), gallic acid, acarbose, α-amylase (procaine pancreas), Dinitrosalicilic acid (DNS), and starch azure were purchased from Sigma-Aldrich (St. Louis, MO, USA). TPTZ (2,4,6-tripyridyl-S-triazine), Iron (III) chloride ($FeCl_3$), 2,2′-azino-bis (3-ethylbenzothiazoline-6-sulfonic acid (ABTS), 2-thiobarbituric acid, 2,2′-Azobis(2-amidinopropane) dihydrochloride (AAPH), Trolox (6-hydroxy-2,5,7,8-tetramethyl chromane 2-carboxylic acid), potassium persulfate ($K_2S_2O_8$), and sodium carbonate ($Na_2CO_3$) were purchased from Sigma-Aldrich (St. Louis, MO, USA). The α-glucosidase (*Saccharomyces cerevisiae*) and p-nitrophenyl-α-D-glucopyranoside (*p*NPG) were obtained from Sisco Research Laboratories Pvt. Ltd. (India).

### Samples collection and preparation

In this study, the indigenous Thai fruits were purchased in the ripe stage. A total of 5 kg of each sample was collected from local markets as shown in Table 1. The samples were dried in an oven at 45 °C for 48 hr and afterward finely ground using a mixer until they reached a powdered consistency. The samples were extracted in triplicate using the following method 6 g of the dried samples were extracted with 80% methanol (50 ml) and sonicated for 30 min at 35 °C using an ultrasonicator (GT SONIC-R3, China). The extracts were filtered through Whatman No. 4 filter paper and the extraction was concentrated using a rotary

**Table 1  Family, scientific name, Thai name and picture of indigenous Thai fruits (the scales of the pictures are unequal).**

| No. | Family | Scientific name | Thai name | Picture |
|---|---|---|---|---|
| 1 | Euphorbiaceae | *Antidesma puncticulatum* | Mao luang | |
| 2 | Dilleniaceae | *Dillenia indica* | Ma tat | |
| 3 | Ebenaceae | *Diospyros decandra* | Chan | |
| 4 | Elaeagnaceae | *Elaeagnus latifolia* | Ma lot | |
| 5 | Salicaceae | *Flacourtia indica* | Ta khop pa | |
| 6 | Clusiaceae | *Garcinia dulcis* | Ma phut | |
| 7 | Sapindaceae | *Lepisanthes fruticosa* | Chamma liang | |
| 8 | Sapotaceae | *Mimusops elengi* | Phikun | |
| 9 | Muntingiaceae | *Muntingia calabura* | Ta khop farang | |
| 10 | Euphorbiaceae | *Phyllanthus reticulatus* | Kang pla khruea | |
| 11 | Moraceae | *Streblus asper* | Khoi | |
| 12 | Myrtaceae | *Syzygium cumini* | Wa | |
| 13 | Myrtaceae | *Syzygium malaccense* | Chompu Mameaw | |
| 14 | Sapindaceae | *Schleichera oleosa* | Ta khro | |
| 15 | Apocynaceae | *Willughbeia edulis* | Katang ka tio | |

evaporator model Büchi Rotavapor® R-210 (Mumbai, India) at 45 °C under a vacuum of 100 mbar. The concentrated extracts were then stored at −20 °C. The extracts were dissolved with 80% methanol for further HPLC and bioactivity analysis.

## Phytochemical evaluation

Total phenolic content (TPC) was measured in triplicate using the Folin-Ciocalteu (F-C) method. Briefly, 30 µL of each extract was mixed with 150 µL of Folin-Ciocalteu reagent (25%, v/v) in a 96-well plate and incubated for 5 min. Then, 120 µL of 10% sodium carbonate was added to the mixture. The mixture was incubated for 60 min at 25 °C in the dark and the absorbance was recorded at 765 nm using the microplate reader model Spark™ 10M (TECAN, Switzerland). The results were given as milligrams of gallic acid equivalents per gram of sample (mg GAE/g) (*Blainski, Lopes & De Mello, 2013*).

The aluminum chloride method analyzed the flavonoid content (TFC) in triplicate using the aluminum chloride method. Briefly, 90 µL of the extract was mixed with 90 µL

of a 2% aluminum chloride solution in a 96-well plate. The mixture was incubated for 15 min at 25 °C, and the absorbance was recorded at 440 nm. The results were presented in milligrams of quercetin equivalents per gram of sample (mg QE/g) (*Molole, Gure & Abdissa, 2022*).

## Identification and quantitative analysis of phenolic compound

The method was modified by *Soto et al. (2022)*. Ultra-high pressure liquid chromatography (UHPLC) was performed on an Agilent 1290 Infinity II LC system (Agilent, Santa Clara, CA, USA), which includes a quaternary solvent pump, an automatic injector, and a column oven. A diode array detector (DAD) was used for analysis. The extracts were separated using a Raptor ARC-18 column (150 mm × 4.6 mm, 2.7 µm particle size; Restek, Centre County, PA, USA). The injection volume was 10 µL and the column was maintained at 40 °C. The mobile phase consisted of a gradient mixture of solvent A (water containing 0.1% $H_2PO_4$) and solvent B (methanol containing 0.1% $H_2PO_4$) with a 0.5 ml/min flow rate. The gradient was started with 90.0% solvent A and 10.0% solvent B and was adjusted to 82.8% A and 17.2% B at 3 min, 77.0% A and 23.0% B at 6.5 min, 68.7% A and 31.3% B at 8.5 min, 54.0% A and 46.0% B at 10 min, 45.0% A and 55.0% B at 11.5 min, 0.0% A and 100.0% B at 13 min, and 90.0% A and 10.0% B at 17 min. The DAD was used at 286 nm. Data acquisition and processing were performed using the Agilent HPLC OpenLAB CDS 2.X software.

### *In vitro* antioxidant assays of extracts

2,2-Diphenyl-1-picrylhydrazyl (DPPH) assay was used to determine the antioxidant activity of the extracts in triplicate. Briefly, 90 µL of the extract was added to 90 µL of methanolic DPPH dye and 90 µL of methanol in 96-well plates, and the reactants of control were prepared by adding 90 µL of methanol to 90 µL of methanolic DPPH dye and 90 µL of methanol in 96-well plates. The mixtures were incubated for 30 min at 25 °C in the dark and the absorbance was measured at 520 nm (*Molyneux, 2003*).

A ferric-reducing antioxidant power (FRAP) assay was used to determine the antioxidant activity of the extracts. Briefly, the FRAP reagent was prepared by mixing a solution of 10 mM TPTZ in 40 mM HCl, acetate buffer (300 mM, pH 3.6), and 20 mM $FeCl_3$ at 10:1:1 (v/v/v). The reactants were prepared by adding 285 µL FRAP reagent to the 15 µL extracts and Trolox (used as a standard) and then incubated for 30 min in the dark at 25 °C. The absorbance was measured at 593 nm. The results were expressed as Trolox equivalents (mm TE). The samples were determined in triplicate (*Fernandes et al., 2016*).

2,2′-azinobis 3-ethylbenzthiazoline-6-sulfonic acid (ABTS) assay was used to determine the antioxidant activity of the extracts. The samples were determined in triplicate. Briefly, the $ABTS^+$ radical was prepared by mixing 2.45 mM $K_2S_2O_8$ and 7 mM ABTS at a 1:1 (v/v) ratio. Then, the mixture was incubated at 25 °C and was kept in the dark for 16 hr. The reactants of the sample were prepared by adding 20 µL of the sample and 180 µL of the $ABTS^+$ radical into 96-well plates; the reactants of control were prepared by adding 20 µL of the methanol and 180 µL of the $ABTS^+$ radical into 96-well plates; The reactants were incubated for 15 min at 25 °C and the absorbance was measured at 734 nm (*Dong et al., 2015*).

The DPPH and ABTS assay results in this study were reported as half-maximal inhibitory concentration values ($IC_{50}$), the concentration of a substance that can inhibit 50% of biological function. The $IC_{50}$ value was calculated in Eqs. (1) and (2). The inhibitory concentration (IC) was calculated by Eq. (1), where the absorbance control represents the absorbance value obtained from the control sample. The absorbance sample represents the absorbance value obtained from the tested sample.

$$\%IC = \left( \frac{\text{Absorbance}_{\text{control}} - \text{Absorbance}_{\text{sample}}}{\text{Absorbance}_{\text{control}}} \right) \times 100\%. \tag{1}$$

Then, to determine the $IC_{50}$, a graph of the relationship between the inhibitory concentration and percent inhibition was made. The regression equation was derived from the graph and the $IC_{50}$ value was calculated by the following equation "a" represents the slope of the dose–response curve, while "b" represents the y-intercept of the dose–response curve.

$$IC_{50} = \left( \frac{50 - b}{a} \right). \tag{2}$$

### *In vitro* antidiabetic assay of extracts

The anti-diabetic activity was quantified using the α-glucosidase inhibition assay as follows: 100 µL of potassium phosphate buffer (10 mM, pH 6.8), 20 µL of α-glucosidase (1U/ml), and 40 µL of extract were mixed into 96-well plates, and afterward incubated for 15 min at 37 °C. Then, 40 µL of 2mM *p*NPG were added and incubated for 15 min at 37 °C. After the incubation, 100 µL of 0.1 M $Na_2CO_3$ were added and the absorbance was measured at 405 nm. A mixture without the extract was used as a blank, while a mixture without the extract and enzyme was taken as a control. Acarbose was used in the assay as a positive control (*Lordan et al., 2013*).

The α-amylase inhibition was determined by the following method: 40 µL of the extract and 40 µL of 1% starch solution were added into a microcentrifuge tube (1.5 ml, Thermo Fisher), incubated for 10 min at 25 °C. Then, 40 µL of α-amylase solution (0.5 mg/ml) was added and incubated for 10 min at 25 °C. Afterward, 80 µL of DNS was added and incubated for 5 min at 100 °C. The reactants were cooled at 0 °C for 5 min. Next, 50 µL of the reactant's solution and 200 µL of DI water were added to 96-well plates. The absorbance was measured at 540 nm. A mixture without the extract was used as a blank, while a mixture without the extract and enzyme was taken as a control. Acarbose was used in the assay as a positive control (*Figueiredo-Gonzalez et al., 2016*).

The inhibition of α-glucosidase and α-amylase was determined using Eq. (1). After determining IC, the $IC_{50}$ values were calculated using Eq. (2).

### Statistical analysis

A two-way analysis of variance was performed in the Jamovi Program (*The Jamovi Project, 2019*) version 0.9.5.12 to determine the effect of indigenous Thai fruit extracts on chemical constituents and bioactivities. *Post-hoc* comparisons between the extracts were performed with Tukey's HSD test. The *p*-values <0.05 were considered statistically significant.

**Table 2  Quantitative analysis of phenolic and flavonoid contents in fruit extracts.**

| No. | Samples | TPC (mg GAE/g) | TFC (mg QE /g) |
|---|---|---|---|
| 1 | *A. puncticulatum* | 81.21 ± 0.62[efg] | 0.03 ± 0.02[fg] |
| 2 | *D. indica* | 80.44 ± 0.20[efg] | 0.94 ± 0.14[bc] |
| 3 | *D. decandra* | 61.03 ± 0.59[fgh] | 0.37 ± 0.5[f] |
| 4 | *E. latifolia* | 48.25 ± 0.59[h] | 0.12 ± 0.01[ce] |
| 5 | *F. indica* | 103.53 ± 0.14[e] | 0.37 ± 0.04[d] |
| 6 | *G. dulcis* | 52.13 ± 0.38[gh] | 1.06 ± 0.08[b] |
| 7 | *L. fruticosa* | 188.19 ± 0.95[c] | 0.77 ± 0.05[c] |
| 8 | *M.s elengi* | 48.47 ± 0.22[h] | 0.13 ± 0.01[ef] |
| 9 | *M. calabura* | 148.63 ± 0.91[d] | 0.33 ± 0.01[d] |
| 10 | *P. reticulatus* | 69.63 ± 0.04[fgh] | 0.01 ± 0.02[fg] |
| 11 | *S. asper* | 75.62 ± 0.37[efgh] | 0.29 ± 0.01[de] |
| 12 | *S. cumini* | 980.42 ± 0.89[a] | 3.55 ± 0.02[a] |
| 13 | *S. malaccense* | 235.98 ± 0.41[b] | 0.36 ± 0.11[d] |
| 14 | *S. oleosa* | 59.67 ± 0.44[fgh] | 0.38 ± 0.03[g] |
| 15 | *W. edulis* | 84.76 ± 0.55[ef] | 0.07 ± 0.04[f] |

**Notes.**
TPC, total phenolic content; TFC, total flavonoid content; GAE, gallic acid equivalent; QE, quercetin equivalent.
Values are mean ± standard deviation in triplicate ($n = 3$). Values in each column with superscript letters (a–d) are significantly different from each other ($p < 0.05$) from Tukey Honest Significant Difference test.

# RESULTS

## Phytochemical evaluation

Methanolic extracts were used for phytochemical evaluation in this study. The total phenolic content was reported as gallic acid equivalents per gram of methanolic extract (mg GAE/g). As shown in Table 2, the highest amounts of total phenolic contents were found in *S. cumini* (980.42 ± 0.89 mg GAE/g), followed by *S. malaccense* (235.98 ± 0.41 mg GAE/g), and *L. fruticose* (188.19 ± 0.95 mg GAE/g), respectively. Furthermore, this study compared the quantity of flavonoids in 15 fruits. As shown in Table 2, the total flavonoid content was reported as quercetin equivalents per gram of methanolic extract (mg QE/g). The highest amounts of flavonoid content were found in *S. cumini* (3.55 ± 0.02 mg QE/g), followed by *E. latifolia* (1.06 ± 0.08 mg QE/g), *D. indica* (0.94 ± 0.14 mg QE/g), and *L. fruticosa* (0.77 ± 0.05 mg QE/g), respectively. These findings indicate that *S. cumini* could be a good source of phenolic and flavonoid supplements compared to all of the fruits in this study. The variation in total phenolic and flavonoid content among samples may be due to genetic factors and ecological conditions. The high phenolic and flavonoid content in *S. cumini* is consistent with previous studies demonstrating its potential health benefits. *S. cumini*, therefore, offers antioxidant, anti-inflammatory, and anti-diabetic properties (*Priya, Prakasan & Purushothaman, 2017*).

## Identification and quantitative analysis of phenolic compound

The methanolic crude extract was analyzed for phenolic compounds through ultra-high-performance liquid chromatography (UHPLC) at a wavelength of 286 nm. The study identified six phenolic compounds (catechin, epicatechin, epicatechin gallate,

**Table 3  Identification and Quantitation of Phenolic compounds.**

| No. | Sample | Catechin (μg/mg) | Ellagic acid (μg/mg) | Epicatechin (μg/mg) | Epicatechin gallate (μg/mg) | Gallic acid (μg/mg) | Kaempferol (μg/mg) |
|---|---|---|---|---|---|---|---|
| 1 | *A. puncticulatum* | ND | ND | ND | ND | 134.57 ± 0.04[h] | ND |
| 2 | *D. indica* | ND | 20.44 ± 0.12[d] | ND | ND | ND | 13.85 ± 0.99[b] |
| 3 | *D. decandra* | 171.54 ± 0.76[d] | ND | ND | ND | ND | 9.45 ± 0.19[c] |
| 4 | *E. latifolia* | ND | 6.02 ± 0.04[e] | 180.28 ± 2.49[c] | ND | 60.86 ± 0.45[i] | 6.89 ± 0.49[cd] |
| 5 | *F. indica* | ND | ND | ND | ND | 46.29 ± 0.07[j] | 16.75 ± 0.11[b] |
| 6 | *G. dulcis* | ND | ND | ND | ND | 552.51 ± 2.99[c] | 24.45 ± 0.95[a] |
| 7 | *L. fruticosa* | 473.79 ± 3.58[c] | ND | ND | ND | 196.16 ± 0.66[g] | ND |
| 8 | *M.s elengi* | ND | 6.91 ± 0.36[e] | ND | 58.14 ± 0.85[b] | 689.26 ± 0.49[b] | ND |
| 9 | *M. calabura* | 93.12 ± 0.49[e] | 89.91 ± 0.63[b] | 1101.8 ± 8.16[b] | ND | 2118.55 ± 6.44[a] | 1.13 ± 0.32[ef] |
| 10 | *P. reticulatus* | ND | 47.65 ± 0.34[c] | ND | ND | 384.87 ± 1.06[f] | 4.69 ± 0.43[de] |
| 11 | *S. asper* | ND | ND | ND | ND | ND | 3.28 ± 0.07[def] |
| 12 | *S. cumini* | 2048.83 ± 1.98[a] | 172.45 ± 0.16[a] | 5397.40 ± 3.03[a] | 3843.07 ± 1.93[a] | 436.44 ± 0.23[e] | 1.40 ± 0.13[ef] |
| 13 | *S. malaccense* | 73.73 ± 0.60[f] | 7.82 ± 0.11[e] | ND | ND | 457.74 ± 0.59[d] | ND |
| 14 | *S. oleosa* | 728.26 ± 3.69[b] | ND | ND | ND | ND | ND |
| 15 | *W. edulis* | ND | ND | ND | ND | 64.61 ± 0.13[i] | 2.59 ± 0.34[ef] |

**Notes.**

Values are mean ± standard deviation in triplicate ($n = 3$). Values in each column with superscript letters (a-d) are significantly different from each other ($p < 0.05$) from Tukey Honest Significant Difference test.

ND, not detected.

gallic acid, kaempferol, and ellagic acid). The results, presented in Table 3, showed that catechin has the highest concentration in *S. cumini* (2048.83 ± 0.98 μg/mg), followed by *S. oleosa* (728.26 ± 0.69 μg/mg), and *L. fruticose* (473.79 ± 0.58 μg/mg). Epicatechin had the high concentration in *S. cumini* (5397.40 ± 0.03 μg/mg), and *M. calabura* (1101.8 ± 0.16 μg/mg). Epicatechin gallate was abundant in *S. cumini* (3843.07 ± 1.93 μg/mg). Ellagic acid was found in high concentrations in *S. cumini* (172.45 ± 0.16 μg/mg), and *M. calabura* (89.91 ± 0.63 μg/ml). Kaempferol was detected in low concentrations in *G. dulcis* (24.45 ± 0.95 μg/ml), *F. indica* (16.75 ± 0.11 μg/mg), and *D. indica* (13.85 ± 0.99 μg/mg). Gallic acid was found in most of the samples, with the highest concentration in *M. calabura* (2118.55 ± 0.44 μg/mg), followed by *M.s elengi* (689.26 ± 0.49 μg/mg), and *G. dulcis* (552.51 ± 0.99 μg/mg).

## Antioxidant capacities of crude extracts

The methanolic extracts were determined for antioxidant activities using three assays: DPPH, FRAP, and ABTS, shown in Table 4. The DPPH assay is a colorimetric reaction that is widely used and easy to perform. The results are expressed as $IC_{50}$ value and indicate that *S. cumini* ($IC_{50}$ value of 3.00 ± 0.01 μg/ml) had the highest antioxidant potential among the compounds tested, followed by *D. decandra* ($IC_{50}$ value of 110 ± 0.04 μg/ml), and *G. dulcis* ($IC_{50}$ value of 120 ± 0.01 μg/ml). The ABTS assay measures the ability of antioxidants to scavenge ABTS radicals generated in aqueous phase. The results are expressed as mg of Trolox and show that *S. cumini* ($IC_{50}$ value of 40 ± 0.01 μg/ml) had the highest antioxidant potential, followed by *S. malaccense* ($IC_{50}$ value of 430 ± 0.02 μg/ml) and *L. fruticose* ($IC_{50}$

**Table 4** Antioxidant activities of indigenous Thai fruits.

| No. | Samples | DPPH (IC$_{50}$μg/ml) | ABTS (IC$_{50}$ μg/ml) | FRAP (mg TE/ml) |
|---|---|---|---|---|
| 1 | *A. puncticulatum* | 1160 ± 0.01[b] | 2200 ± 0.08[c] | 169.41 ± 0.69[c] |
| 2 | *D. indica* | 690 ± 0.01[de] | 2240 ± 0.08[e] | 4.99 ± 0.45[e] |
| 3 | *D. decandra* | 110 ± 0.04[g] | 1870 ± 0.01[c] | 6.79 ± 0.09[e] |
| 4 | *E. latifolia* | 1060 ± 0.16[bc] | 4750 ± 0.07[b] | 6.96 ± 0.08[e] |
| 5 | *F. indica* | 140 ± 0.01[g] | 650 ± 0.01[c] | 17.23 ± 0.23[e] |
| 6 | *G. dulcis* | 120 ± 0.01[g] | 2130 ± 0.40[c] | 2.68 ± 0.06[e] |
| 7 | *L. fruticosa* | 740 ± 0.02[de] | 500 ± 0.06[e] | 19.45 ± 0.81[e] |
| 8 | *M.s elengi* | 620 ± 0.27[de] | 1170 ± 0.06[d] | 0.91 ± 0.61[e] |
| 9 | *M. calabura* | 550 ± 0.01[def] | 610 ± 0.02[e] | 27.41 ± 0.23[e] |
| 10 | *P. reticulatus* | 330 ± 0.04[fg] | 1420 ± 0.27[d] | 6.1 ± 0.06[e] |
| 11 | *S. asper* | 630 ± 0.25[de] | 2260 ± 0.18[c] | 9.23 ± 0.42[e] |
| 12 | *S. cumini* | 3.00 ± 0.01[a] | 40 ± 0.01[a] | 898.63 ± 0.02[a] |
| 13 | *S. malaccense* | 210 ± 0.02[g] | 430 ± 0.02[e] | 484.75 ± 0.66[b] |
| 14 | *S. oleosa* | 820 ± 0.14[cd] | 2150 ± 0.02[c] | 11.29 ± 0.06[e] |
| 15 | *W. edulis* | 520 ± 0.01[ef] | 2110 ± 0.10[c] | 124.93 ± 0.77[d] |

**Notes.**
Values are mean ± standard deviation in triplicate ($n = 3$). Values in each column with superscript letters (a–d) are significantly different from each other ($p < 0.05$) from Tukey Honest Significant Difference test.

value of 500 ± 0.06 μg/ml). The FRAP assay measures the antioxidant capacity by reducing ferric ions to ferrous ions, and the results are expressed as $Fe^{2+}$ equivalents or FRAP values. The results revealed that *S. cumini* (898.63 ± 0.02 mg TE/ml) had the highest antioxidant potential, followed by *S. malaccense* (484.75 ± 0.66 mg TE/ml), and *A. puncticulatum* (169.41 ± 0.69 mg TE/ml). Overall, the results indicate that *S. cumini* and *S. malaccense* are excellent sources of antioxidant compounds.

## Antidiabetic activities of crude extracts

The antidiabetic capacity of methanolic extracts was determined using two key enzyme assays. Including α-glucosidase inhibition and α-amylase inhibition. The α-glucosidase inhibition assay measures the potential of antidiabetic activity and the results are expressed as IC$_{50}$. The results, shown in Table 5, reveal that *S. cumini* (IC$_{50}$ value of 0.13 ± 0.01 mg/ml) had the highest potential of α-glucosidase inhibition among the samples tested, followed by *M. calabura* (IC$_{50}$ value of 3.27 ± 0.82 mg/ml), and *D. decandra* (IC$_{50}$ value of 3.96 ± 0.19 mg/ml). Additionally, Acarbose was included as a positive control, exhibiting an IC$_{50}$ value of 0.86 ± 0.01 mg/ml. The α-amylase inhibition assay is also used to measure the potential of antidiabetic activity and the results are expressed as IC$_{50}$values. The results, shown in Table 5, emphasize that *S. cumini* (IC$_{50}$ value of 3.91 ± 0.05 mg/ml) had the highest ability of α-amylase inhibition, followed by *L. fruticosa* (IC$_{50}$ value of 4.14 ± 0.04 mg/ml), and *W. edulis* (IC$_{50}$ value of 4.88 ± 0.02 mg/ml). Acarbose, the positive control, exhibited an IC$_{50}$ value of 0.39 ± 0.05 mg/ml. Overall, the results indicate that *S. cumini* has the highest potential for antidiabetic activity among the samples tested.

**Table 5** Antidiabetic activities of indigenous Thai fruits.

| No. | Samples | $\alpha$-Glucosidase inhibition IC$_{50}$ (mg/ml) | $\alpha$-amylase inhibition IC$_{50}$ (mg/ml) |
|---|---|---|---|
| 1 | *A. puncticulatum* | 42.76 ± 1.08[c] | 5.51 ± 0.03[e] |
| 2 | *D. indica* | 62.65 ± 1.86[b] | 17.90 ± 0.07[c] |
| 3 | *D. decandra* | 3.96 ± 0.19[f] | 28.84 ± 0.05[b] |
| 4 | *E. latifolia* | 95.52 ± 0.53[a] | 46.6 ± 0.22[a] |
| 5 | *F. indica* | 26.23 ± 0.08[f] | 11.38 ± 0.14[d] |
| 6 | *G. dulcis* | 12.54 ± 0.28[f] | 15.71 ± 0.36[d] |
| 7 | *L. fruticosa* | 5.70 ± 0.20[f] | 4.14 ± 0.04[e] |
| 8 | *M.s elengi* | 13.01 ± 0.64[f] | 7.18 ± 0.02[e] |
| 9 | *M. calabura* | 3.27 ± 0.82[f] | 13.89 ± 0.14[d] |
| 10 | *P. reticulatus* | 30.21 ± 3.29[de] | 5.18 ± 0.01[e] |
| 11 | *S. asper* | 60.40 ± 1.23[b] | 24.72 ± 0.09[b] |
| 12 | *S. cumini* | 0.13 ± 0.01[e] | 3.91 ± 0.05[e] |
| 13 | *S. malaccense* | 54.43 ± 2.06[b] | 5.82 ± 0.04[e] |
| 14 | *S. oleosa* | 13.42 ± 0.34[f] | 5.25 ± 0.04[e] |
| 15 | *W. edulis* | 39.39 ± 1.36[cd] | 4.88 ± 0.02[e] |
| 16 | Acarbose | 0.86 ± 0.01 | 0.39 ± 0.05 |

**Notes.**

The values provided in the tables are the mean values obtained from triplicate measurements, with the standard deviation also provided. The values in each column with superscript letters (a–d) are statistically significant from one another, as determined by the Tukey Honest Significant Difference test ($p < 0.05$).

## DISCUSSION

This study, investigated the total bioactive content encompassing phenolic and flavonoid content, we used three different radical scavenging assays to analyze the antioxidant abilities of various fruit extracts. The assays included the DPPH assay and the measurement of the sample's ability to scavenge DPPH radicals. DPPH radicals are soluble in organic media and thus, DPPH is commonly used to screen for bioactive compounds such as phenols and flavonoids (*Gulcin & Alwasel, 2023*), the ABTS assay measures the sample's ability to scavenge ABTS radical cations. ABTS radicals are soluble in organic and aqueous mediums, allowing them to screen for lipophilic and hydrophilic samples. The FRAP assay measures the reducing power of the sample (*Sadeer et al., 2020*). We have chosen these three assays to ensure the reliability of our results. We also analyzed antidiabetic activity by measuring the inhibition of two key enzyme activities: α-amylase, which breaks down complex carbohydrates into smaller polysaccharides, and α-glucosidase, which breaks down disaccharides and oligosaccharides into glucose that can be absorbed by the human body (*Li et al., 2022*). All plant extracts exhibited antioxidant activity in all three assays and antidiabetic activity in both enzyme assays. In particular, *S. cumini* showed prominent antioxidant and antidiabetic activities and had the highest total phenolic content and flavonoid content among all samples. The correlation analysis between Total Flavonoid (TFC), Total Phenolic Content (TPC), antioxidant properties (FRAP, ABTS, DPPH), and antidiabetic properties (α-amylase inhibition, α-glucosidase inhibition)

with Pearson's correlation test reveals several significant relationships. Notably, TFC is strongly positively correlated with FRAP ($r = 0.97$) and ABTS ($r = 0.76$), while TPC is moderately correlated with α-amylase inhibition ($r = 0.53$). However, TPC shows strong negative correlations with TFC ($r = -0.96$) and FRAP ($r = -1$), possibly because the total phenolic content in plants can vary significantly, with non-flavonoid phenolics potentially being more predominant and contributing to higher total phenolic content (*John et al., 2016*), and a moderate negative correlation with ABTS ($r = -0.53$). These findings are consistent with previous research, showing that specific structural features in flavonoids, such as hydroxyl groups and double bonds, can enhance their antioxidant and antidiabetic properties (*Ahmed et al., 2018*; *Sarian et al., 2017*). Furthermore, higher intake of total flavonoids has been associated with a decreased risk of developing type 2 diabetes mellitus (*Xu et al., 2018*). Additionally, this research conducted a comparative analysis of various commercial fruits through an extensive review of the existing literature, which used a similar extraction method. The assessment of antioxidant properties, as measured by the DPPH assay, revealed that the samples examined in this study, which included *S. cumini, D. decandra, F. indica, S. malaccense,* and *P. reticulatus*, exhibited superior antioxidant properties when compared to well-known fruits such as *Punica granatum* (Pomegranate, DPPH IC$_{50}$ $0.32 \pm 0.01$ mg/ml), *Malus domestica* (Apple, DPPH IC$_{50}$ $1.65 \pm 0.04$ mg/ml), *Prunus armeniaca* (Apricot, DPPH IC$_{50}$ $1.67 \pm 0.03$ mg/ml), *Citrus reticulata* (Mandarin, DPPH IC$_{50}$ $4.92 \pm 0.09$ mg/ml), and *Prunus persica* (Peach, DPPH IC$_{50}$ $0.98 \pm 0.02$ mg/ml) (*Habiba, Seddik & Amel, 2020*). Furthermore, this research involved a comparative analysis of the antidiabetic capabilities of the studied fruits. Notably, *S. cumini* in this study demonstrated superior α-amylase and α-glucosidase inhibition compared to commercially known fruits, including *Mangifera indica* (mango, α-amylase inhibition; IC$_{50}$ $0.287$ mg/ml and α-glucosidase inhibition; IC$_{50}$ $112.8$ mg/ml) (*Sekar et al., 2019*), *Citrus macroptera* (wild orange, α-amylase inhibition; IC$_{50}$ $3.638 \pm 0.19$ mg/ml) (*Uddin et al., 2014*), *Malus domestica* (α-amylase inhibition; IC$_{50}$ $0.25$ mg/ml) (*Utami et al., 2019*), *Prunus armeniaca* (α-amylase inhibition; IC$_{50}$ $1.30 \pm 0.02$ mg/ml) (*Kaya & Keski, 2021*), and *Prunus persica* (α-amylase inhibition; IC$_{50}$ $3.24 \pm 0.05$ mg/ml and α-glucosidase inhibition; IC$_{50}$ $7.20 \pm 0.20$ mg/ml) (*Nowicka et al., 2023*).

Based on the preceding results regarding total phenolic and flavonoid content, the next investigation focuses on identifying the specific phenolic compound in the crude extract. The study identified and quantified phenolic compounds in crude extracts. Catechins (including catechin, epicatechin, and epicatechin gallate) were abundant in *S. cumini*, while epicatechin was abundant in *M. calabura*. Gallic acid was found in most samples and was particularly abundant in *M. calabura*. Kaempferol was present in small amounts in most samples, and ellagic acid was found in low amounts in some of the samples analyzed. Due to the results, we expected that catechins might be one of the powerful active compounds for antioxidant and antidiabetic activities. Likewise, studies have shown that catechins have a powerful antioxidant activity by scavenging free radicals. Potential antidiabetic inhibition can be achieved by reducing reactive oxygen species by suppressing NADPH oxidase activity (*Mrabti et al., 2018*). Improving mitochondrial function causes insulin release, increasing the inhibition of blood glucose. Furthermore, an improvement in intestinal

function and high anti-inflammatory activity can be noticed (*Wen et al., 2022*). Gallic acid was reported as a powerful antioxidant and antidiabetic agent (*Salih, 2010*) . Kaempferol has been demonstrated to effectively inhibit α-glucosidase activity, thereby regulating glucose levels in the body (*Pereira et al., 2011*). Additionally, another study confirmed its anti- α-glucosidase properties. The results indicate that kaempferol, with its lower $IC_{50}$ value, is a more potent α-glucosidase inhibitor than quercetin (*Yulia et al., 2020*). Ellagic acid has been reported for its antioxidant ability through the scavenging of reactive oxygen species it increases the expression of antioxidant enzymes such as superoxide dismutase, glutathione peroxidase, glutathione reductase, and catalase (*Sharifi et al., 2022*). Another study, it was discovered that kaempferol led to a dose-dependent increase in serum insulin levels in diabetic rats (*Fatima et al., 2017*). In addition, the increase of blood glucose causes oxidative stress in β-cell and leads to dysfunction, apoptosis, and necrosis of β-cell. This affects insulin secretion and function which can lead to diabetes. Therefore, increased free radical scavenging agents can lower the risk of diabetes and alleviate its symptoms (*Sun et al., 2021*).

## CONCLUSION

This research focused on both the antioxidant and antidiabetic activities, and the phytochemical evaluation of various samples. For the phytochemical evaluation, methanolic extracts were used; the highest total phenolic contents were found in *S. cumini*, followed by *S. malaccense*, and *L. fruticose*, respectively. The highest amounts of flavonoids were found in *S. cumini,* followed by *E. latifolia*, *D. indica*, and *L. fruticosa*. It was found that *S. cumini* could be considered a good source of phenolic and flavonoid supplements, compared to other fruits in this research. Three assays were used to measure the antioxidant capacities of crude extracts: DPPH, FRAP, and ABTS. The results revealed that *S. cumini* has the highest antioxidant potential among the compounds tested. The antioxidant activities of *S. cumini* and *S. malaccense* positively correlate to their total phenolic content. Two assays were used for antidiabetic activities of crude extracts: α-glucosidase inhibition and α-amylase inhibition. The results showed that *S. cumini* has the highest potential for α-glucosidase and α-amylase inhibition among the samples tested, indicating that it has the highest potential for antidiabetic activity. This study involves a preliminary assessment of antioxidant and antidiabetic activities in crude extracts. We propose further fractionation and purification of the extract to enhance bioactivities, pinpointing the active compound responsible for these effects. Moreover, we recommend conducting *in vivo* and clinical tests to validate these findings for future research.

## ACKNOWLEDGEMENTS

The authors thank the Natural Products Extraction and Isolation Laboratory, Department of Medical Sciences for providing the equipment. Special thanks are also extended to Mr. Aussavashai Shuayprom and Mr. Stephan Zentner for valuable advice.

### Funding

This research is funded by Kasetsart University through the Graduate School Fellowship Program. The funders had no role in study design, data collection and analysis, decision to publish, or preparation of the manuscript.

### Grant Disclosures

The following grant information was disclosed by the authors:
Kasetsart University through the Graduate School Fellowship Program.

### Competing Interests

The authors declare there are no competing interests.

### Author Contributions

- Jirayupan Prakulanon conceived and designed the experiments, performed the experiments, analyzed the data, prepared figures and/or tables, authored or reviewed drafts of the article, and approved the final draft.
- Sutsawat Duangsrisai analyzed the data, authored or reviewed drafts of the article, and approved the final draft.
- Srunya Vajrodaya analyzed the data, authored or reviewed drafts of the article, and approved the final draft.
- Thanawat Thongchin analyzed the data, authored or reviewed drafts of the article, and approved the final draft.

### Data Availability

Raw data, including HPLC and bioactivities analysis, are available in the Supplemental Files.

### Supplemental Information

Supplemental information for this article can be found online at http://dx.doi.org/10.7717/peerj.17681#supplemental-information.

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
