# Peer review of "Evaluation of phytochemical profile, and antioxidant, antidiabetic activities of indigenous Thai fruits"

_PeerJ, doi:10.7717/peerj.17681_

## Round 0.1 · original submission · Major Revisions

Please respond to all reviewer comments with a point-by-point rebuttal letter and submit a revised version.

**Language Note:** The review process has identified that the English language must be improved. PeerJ can provide language editing services - please contact us at [email protected] for pricing (be sure to provide your manuscript number and title). Alternatively, you should make your own arrangements to improve the language quality and provide details in your response letter. – PeerJ Staff

Reviewer 1 ·

Basic reporting

1. Literature references: The reference by Chassagne et al. on line 44 needs verification as it was not found, including the publication year.
2. The introduction: it needs to incorporate a discussion on the previous study regarding the active compounds of indigenous Thai fruits and their potential impact on antidiabetic activity, particularly in the context of antioxidant. This addition will help establish a stronger link between your research and the existing literature.
3.The introduction is too brief and uninformative. The authors should include why they were interested in indigenous Thai fruits focusing on its antioxidant and anti-diabetic effect.

Experimental design

1. In materials & methods: In order to enhance the comprehensiveness of this study, please provide details regarding the static testing method.

Validity of the findings

1. In the results: lines 164-165 should be revised delete references.
2. line number 170, you should verify the plants species, as the reported results do not correspond with every table displaying the experimental outcomes.
3. Line number 220, you should be recheck the number of the table.
4. In the results of table 3 Antidiabetic activities of indigenous Thai fruits. Additionally, it presents the standard values for comparing Acarbose substances.

Additional comments

1. Discussion and conclusion: In the discussion section, elaborate on the specific bioactive compounds that may play a crucial role in the observed antioxidant and antidiabetic activity of S. cumini.
Additionally, discuss the potential mechanisms through which these bioactive exert their effects.
2. Experimental method: Verify accuracy with lines 106-107. Did you confirm the presence of phenolic and flavonoid compounds using UHPLC-DAD.
3. The value should be displayed, depicting the chromatogram of the substance obtained from the UHPLC-DAD analysis technique.

Annotated reviews are not available for download in order to protect the identity of reviewers who chose to remain anonymous.

Reviewer 2 ·

Basic reporting

Minor revision
1. Author may add some more information about why choose 6 kinds of fruit? Do those fruit have been reported or used as traditional medicine?
2. Author may add the photo of fruit in supplementary.
3. Author should add Phenolic compounds std. in material and method.

Experimental design

No comment

Validity of the findings

No comment

·

Basic reporting

The manuscript lacks professional English. Therefore, authors should seek a Professional grammar expert.
However, the article shows a relevant and well-structured research report, with clear and precissed formatted tables.

Experimental design

Except for minor observed and indicated mistakes in the manuscript, no further comments are discussed (see attached file).

Validity of the findings

No further comments.

Additional comments

Original research report showed the antioxidant and antidiabetic potential of Thai fruits, which may be of interest for international academic or industry (e.g., nutraceuticals or pharmaceuticals) colleagues.
However, few changes should be considered before publication. It is strongly suggested to have the manuscript sent for Grammar review and correction by a Professional Expert (See PeerJ for more details).
Also, I encourage authors to further research the hydrophilic Thai fruit extracts for in vivo antidiabetic assays (pre- or clinical interventions).
All suggested and required changes are described in the attached file.

Reviewer 4 ·

Basic reporting

The introduction will be more informative if it is expanded or adjusted to include the following points:
1. Adding the review of fruits with antidiabetic properties and their chemical constituents:
2. Adding the review of the occurrence of Quercetin (Q) and Rutin (R) in fruits and their antidiabetic effects:
3. Adding the review of the other flavonoids with antidiabetic properties:
4. All information presented in this review will be supported by relevant references from scientific studies. This will ensure the accuracy and reliability of the information provided.
5. The rationale behind selecting two specific enzymes for testing in the context of antidiabetic research will be clearly explained.

Experimental design

The comments as follow.
- Chemical and reagent
1. The country of origin of the chemicals should be specified.
2. Information should be provided for each test, including the method of preparing the solutions.
3. The chemical formulas should be checked, e.g., Fe(III)Cl3.6H2O on line 141 and Na2CO3 on line 141.
4. When using abbreviations, the full name should be written first, e.g., ABTS.
5. The source and preparation of the enzymes used should be specified.

- Samples collection and preparation
1. How do you know when the sampling of indigenous Thai fruits is complete and represented? And what are the criteria for sampling?
2. The sample preparation method should be described in detail. For example, the sample preparation method should be referenced, the conditions for grinding the sample should be explained, the brand and model of the sonicator should be specified, the approximate room temperature in this experiment should be given, and the brand and model of the evaporator should be provided.

- Phytochemical evaluation
1. Specify the instrument for measuring absorbance. What brand and model are used?"

- Antioxidant capacities…
1. Provide the full name of the word "DPPH" on line 121.
2. The DPPH method used in the test should be referenced.
3. Abbreviations should be specified in parentheses on lines 125 and 130.
4. Line 126: "The samples were prepared by adding..." should be changed to "The reactants were prepared by adding..." because this step is a reaction step.

- In vitro antidiabetic of extracts
1. After the abbreviation is shown, it can be used throughout the document and should be consistent, for example, pPNG.
2. Specify the equation number and refer to it in the text.
3. Specify the referencing method for Antidiabetic assays.
4. Further explanation of the preparation of iodine reagent solution is needed. The words in parentheses "5 mM iodine and 5 mM potassium iodide" do not convey the preparation.
5. The equations in lines 158 and 159 should be explained in the text. The expression of both enzymes' activities should be displayed in the same calculation.

Validity of the findings

- TPC and TFC
1. Check all the table number references. You should also make sure that all of the tables are numbered in sequence.
2. If you are going to use the abbreviations TPC or TFC, you must state them from the first time they are used, and then use the abbreviations throughout.

- Phenolic compound
1. Explain a reason why the selected group of compounds was chosen for this study since the introduction mentions Quercetin) and rutin.

- Antioxidant
1. Please describe how to calculate the IC50 values for their experiments and specify the units of IC50.
2. Line 201: It is recommended to show the value of the positive correlation coefficient to provide additional evidence to support this information.

- Anti-diabetic activity
1. specify the units of IC50.

Discussion
1. The information in Lines 226-232 must move to the introduction part.
2. The information in Lines 235-236 should have a reference. The link between organic medium and the testing of the phenol and flavonoid groups. The authors should provide a citation for this study so that readers can find more information about it.
3. The fruits mentioned in lines 253-256 should have their various values displayed for comparison of the experimental results, such as antioxidant, TPC, TFC, and antidiabetic values.
4. The text in lines 264-275 should be moved to the introduction section and must be cited.
5. The data from lines 275-279 should be moved to support the findings from the antioxidant activity analysis. This would help to make the findings clearer and more convincing.

Additional comments

-

---

## Round 0.2 · Minor Revisions

Please address the final minor issues indicated by the reviewer #4.

Reviewer 1 ·

Basic reporting

no comment

Experimental design

no comment

Validity of the findings

no comment

Reviewer 2 ·

Basic reporting

no comment

Experimental design

no comment

Validity of the findings

no comment

Additional comments

Authors had revised and add some more detail as suggested, accept in this version

·

Basic reporting

Manuscript has been improved by authors as requested changes and other comments were completely revised and corrected.

Experimental design

The manuscript experimental design was done and executed properly. No further comments.

Validity of the findings

After all manuscript corrections, I find it interesting for both Academy and Industry areas.

Additional comments

Manuscript has been improved by authors as requested changes and other comments were completely revised and corrected.

Reviewer 4 ·

Basic reporting

- Lin 6: Check “botany”
- Line 26-29: “Additionally, UHPLC-DAD was utilized to identify and quantify phenolics and assess antioxidant and antidiabetic abilities”. It is not clear whether UHPLC is used to assess antioxidants and antidiabetic abilities, including α-glucosidase and α-amylase inhibition or other method.
- line Line 62: These phenolics what do you mean?
- line 85: not found reference of Suree et al., 2018, Dedvisitsakul et al., 2022

Experimental design

- line 121-122: The number of samples per Thai fruit must be reported to show the representative sample.
- please explain more about how to prepare a sample solution from the crude extract before taking it to use in each analysis.

Validity of the findings

- Line 152-156: Please check the consistency of the used digital place of %ratio.
- Line 181-182: Please rearrange the sentence “The IC50 value was calculated as follows. The inhibitory concentration (IC) was calculated by this Equation.”
- line 182 &188: Equations 1 and 2 should be informed in text.
- In vitro antioxidant assays: please inform the preparation of the control solution used in each analysis
- Line 190: Please define terms a and b in Equation 2
- Line 230: Please check “quercitin”
- Missing the discussion of correlations between the chemical constituents and bioactivities examined with Pearson's correlation test.

Additional comments

-

---

## Round 0.3 · accepted · Accept

This is Accepted